# Exploring bidirectional causality between religion and mental health: A longitudinal study using data from the parent generation of a UK birth cohort

Daniel Major-Smith [1,2*], Jimmy Morgan[1], Isaac Halstead[1], Jean Golding[1]

**1** Centre for Academic Child Health, Population Health Sciences, Bristol Medical School, University of Bristol, Bristol, United Kingdom, **2** Department of the Study of Religion, Aarhus University, Aarhus, Denmark

* dan.smith@bristol.ac.uk

## Abstract

Relations between religion and mental health have been studied extensively, yet whether associations are causal remains uncertain. Here, we use longitudinal data from the parental generation of the Avon Longitudinal Study of Parents and Children (ALSPAC), based in the UK, to assess: i) whether religiosity may cause subsequent depression and anxiety; ii) whether depression and anxiety may cause subsequent religiosity; and iii) whether there are gender differences in the above associations. All analyses were pre-registered, and adjusted for baseline confounders, exposures and outcomes in an attempt to rule out reverse causality and confounding bias. We found little conclusive evidence that religiosity was associated with subsequent mental health, or that mental health was associated with subsequent religiosity. Some weak associations were reported, but effect sizes were small and largely consistent with null effects. Small differences by gender were found, with religiosity marginally associated with better mental health in women and worse mental health in men, but the inconsistency of the results and the wide margins of error mean that firm conclusions cannot be made. In sum, in this UK population we find little evidence for bidirectional causation between religion and mental health, or for large differences in these associations by gender.

## Introduction

Mental health problems, specifically depression and anxiety, have large personal and societal impacts and are a key cause of disability, morbidity and premature death worldwide [1,2]. Worryingly, their prevalence has also increased over recent decades [3,4]. Understanding the factors that shape mental health onset, duration, recovery and recurrence is therefore of utmost significance. While acknowledging the importance of well-established risk factors, such as genetics, family history, adverse life events, socioeconomic deprivation and co-morbidities [5,6], in this paper we focus on the potential role of religious/spiritual beliefs and behaviours (RSBB) and whether they may plausibly cause depression and anxiety.

**Data availability statement:** ALSPAC data access is through a system of managed open access. Information about access to ALSPAC data is given on the ALSPAC website (http://www.bristol.ac.uk/alspac/researchers/access/) and in the ALSPAC data management plan (http://www.bristol.ac.uk/alspac/researchers/data-access/documents/alspac-data-management-plan.pdf). Data used for this submission will be made available on request to the Executive (alspac-exec@bristol.ac.uk). The datasets presented in this article are linked to ALSPAC project number B4226, please quote this project number during your application. Analysis code and synthetic ALSPAC datasets (created using the 'synthpop' R package; https://www.synthpop.org.uk/), are openly-available on DM-S's GitHub page: https://github.com/djsmith-90/AnalysisCode_RSBBandMH_B4226. As raw ALSPAC data cannot be released, these synthesised datasets are modelled on the original data, thus maintaining variable distributions and relations among variables (albeit not perfectly), while at the same time preserving participant anonymity and confidentiality, thus allowing this research to be 'quasi-reproducible'. Please note that while these synthetic datasets can be used to follow the analysis scripts, as data are simulated they should not be used for research purposes; only the actual, observed, ALSPAC data should be used for formal research and analyses reported in published work.

**Funding:** The UK Medical Research Council and Wellcome Trust (Grant ref: 217065/Z/19/Z) and the University of Bristol currently provide core support for ALSPAC. This publication is the work of the authors, and Daniel Major-Smith will serve as guarantor for the contents of this paper. A comprehensive list of grants funding is available on the study website (http://www.bristol.ac.uk/alspac/external/documents/grant-acknowledgements.pdf). This project was made possible through the support of a grant from the John Templeton Foundation (ref no. 61917). The opinions expressed in this publication are those of the author(s) and do not necessarily reflect the views of the John Templeton Foundation. The funders had no role in study design, data collection and analysis, decision to publish, or preparation of the manuscript.

**Competing interests:** The authors have declared that no competing interests exist.

There are numerous theoretical reasons why religion may protect against developing mental health problems [7,8]. Religion can help provide meaning in life, which may mitigate feelings of hopelessness and anxiety. Religion may also aid in coping with stressful and traumatic life events (e.g., "all in God's plan"), although religion may also be detrimental in this regard if individuals feel that God has abandoned or is punishing them [9,10]. Additionally, religious groups can also provide a sense of community, combating loneliness and providing social support, especially in times of need. On theoretical grounds, we may therefore predict that, overall, religion will cause better mental health.

The literature on religion and mental health is extensive. Taking depression first, in a review of 413 observational studies up to 2010, the majority (61%) reported a protective association between religion and depression [7]. However, most of these observational studies were cross-sectional (83%), making a causal interpretation difficult to establish, especially given the potential for reverse causation; that is, mental health causing changes in religion [11,12]. Results for anxiety are comparable, although generally weaker than for depression; up to 2010, of 299 studies 49% reported a protective association between RSBB and anxiety, although 94% of these studies were cross-sectional [7].

In order to make stronger causal claims from observational studies, data need to be longitudinal, with information on a range of baseline confounders, prior outcomes, and ideally prior exposures as well [13,14]; such designs can help reduce the risk of both confounding bias and reverse causation. Studies using such longitudinal designs which control for baseline mental health have nonetheless replicated these overall patterns of results for depression, with religion appearing protective against later depression and facilitating recovery from depression [12,15–20], albeit not universally [21,22]. Systematic reviews and meta-analyses of longitudinal papers investigating religion and depression have found an overall protective association, although with considerable heterogeneity between studies [23,24]. Longitudinal research on RSBBs and anxiety have generally found associations to be much weaker relative to depression [20], often null [21,25], and occasionally positive, with religiosity sometimes associated with greater anxiety [26].

Despite this work, detailed, large-scale, longitudinal population-based studies on this topic are still relatively rare, and many open questions remain. For instance, the majority of previous studies – 75%, according to a recent meta-analysis [24] – were based in the US. However, as the US is more religious than other, more secular, countries, such as the majority of Europe [27], these findings may be limited in their generalisability [22]. While fewer in number, some longitudinal studies in Europe [19], Canada [17,28] and South Korea [18] have nonetheless reported similar protective associations between religion and subsequent depression. However, other non-US longitudinal studies have reported little association between RSBB and depression, with some heterogeneity between countries, such as a *positive* (i.e., harmful) association reported in the UK [22]. There is therefore a need to conduct studies using longitudinal data in a range of populations to explore how replicable and generalisable these patterns are, and – ultimately – to understand the reasons for any cross-cultural variability.

Additionally, much previous work – over half [24] – has focused on religious attendance rather than other aspects of religiosity. One consequence of this is that it ignores the multidimensional nature of religiosity, which includes not just behaviours (such as religious attendance) but also belief, identity and belonging [29]. A further consequence is that focusing on attendance potentially overlooks important differences between those who do not regularly attend a place of worship, yet have different beliefs about religion, such as between atheists, agnostics, and non-attending religious individuals [30,31]. Where studies have explored alternative aspects of religiosity, associations are generally weaker compared to religious

attendance [16,24–26,28,32]. This suggests that religious attendance may shape behaviour to a greater extent than other facets of religiosity, although the mechanisms remain poorly-understood [32]. Additional research is required to confirm and further investigate these patterns, specifically by exploring the relationship between religiosity and mental health using a range of RSBB variables.

A further open question is the extent to which mental health issues cause RSBB. This has been explored by fewer studies, although existing research has found that depression and anxiety are associated with lower rates of subsequent religious attendance [11,12,28]. This is important from a methodological perspective, as if changes in mental health cause subsequent changes in religiosity, then cross-sectional studies investigating religion and mental health are at risk of reverse causality and may be unable to estimate their causal effect of interest without bias [14]. From theoretical and practical perspectives, this potential bidirectional causation can feed into theories of how religion and mental health interact, as well as help inform potential interventions to improve mental health. For instance, a common symptom of depression is withdrawal from usual social activities, which may include religious activities [11]; conversely, it is also possible that mental health issues could result in an increase in religious coping [33], and hence greater subsequent religiosity, although most longitudinal studies so far have reported a negative relationship between mental health problems and subsequent religiosity. We will therefore also investigate this under-explored topic here to assess whether there is potential bidirectional causation between RSBB and mental health.

Finally, some studies have also noted gender-specific associations between religion and mental health [28]. In general, there appears to be a stronger protective association between religion and depression among women, with many studies reporting null [15], or even positive/harmful [34], associations among men. Associations between depression and subsequent attendance have also been reported for women, but not men [11]. One potential reason for this is gender differences in coping, with women generally more likely to engage in religious coping in times of stress [35] (although see [36]). As women are also more likely to use social support as a method of coping [35] – which could include support from co-religionists – this may also play a role in any gender differences. However, as the number of longitudinal studies exploring such gender differences in religion and mental health is small, there is a need for more research in this area.

We will explore these questions using data from the parental generation of a large-scale population-based UK birth cohort (ALSPAC; the Avon Longitudinal Study of Parents and Children), which has detailed and repeated mental health [37,38] and RSBB [39,40] data in both mothers and their partners. Our study aims to answer the following research questions: i) Does RSBB cause anxiety and depression? ii) Do depression or anxiety cause RSBB? and iii) Are there gender differences in the above associations? Based on theory and previous research, we predict that:

1) RSBB will be protective against developing later depression and anxiety.

2) Associations between RSBB and depression will be stronger than those between RSBB and anxiety.

3) Associations will be stronger for religious attendance, compared to other aspects of RSBB (i.e., religious belief or religious affiliation).

4) Higher levels of depression and anxiety will be associated with lower rates of subsequent religious engagement and attendance.

5) The above associations will be stronger among women compared to men.

## Methods

An analysis plan for all analyses reported in this paper was pre-registered on the Open Science Framework (OSF) website (https://osf.io/qtdze/). The research questions, methods and analyses reported below are identical to those specified in the analysis plan, other than some superficial updates and corrections (see section S1 of the Supporting Information for full details).

### ALSPAC study description

Pregnant women resident in Bristol and surrounding areas, UK, with expected dates of delivery between 1st April 1991 and 31st December 1992 were invited to take part in the study. The initial number of pregnancies enrolled was 14,541, of which there were a total of 14,676 foetuses, resulting in 14,062 live births and 13,988 children who were alive at 1 year of age [41,42]. Of these 14,541 initial pregnancies, there were 14,203 unique mothers [43]. The current research focuses on the parental ALSPAC generation, which includes both the study mothers and their partners. For this study, we removed one pregnancy if the mother had two pregnancies enrolled in ALSPAC (to avoid repeated data from the same parent). We also dropped observations for participants who had withdrawn consent for their data to be used and removed observations where the pregnancy did not result in a live birth (as there would be no post-pregnancy data for these participants). The final dataset contained 13,678 mothers.

For each mother, we also included their associated partner, usually the father of the study child. Initially, partners/fathers (hereafter 'partners') were not formally enrolled into ALSPAC, but were given partner-based questionnaires by the mother (if she had a partner and chose to invite them). This means that partner-based questionnaires may not have been completed by the same partner over time (although numbers of such cases are likely to be relatively small; approx. 5% of all partners); to remove this source of bias, we excluded partners known to have changed identity over the course of the study. In total, 12,113 G0 partners have been in contact with the study by providing data and/or formally enrolling when this started in 2010 (with 3,807 partners currently enrolled [44]). The final dataset contained 13,296 partners.

Please note that the study website contains details of all the data that is available through a fully searchable data dictionary and variable search tool:

http://www.bristol.ac.uk/alspac/researchers/our-data/. Ethical approval for the study was obtained from the ALSPAC Ethics and Law Committee and the Local Research Ethics Committees. Informed consent for the use of data collected via questionnaires and clinics was obtained from participants following the recommendations of the ALSPAC Ethics and Law Committee at the time. For the questionnaire data used in the present study, consent was implied via the completion and returning of questionnaires (for more information on ALSPAC ethics, see [45]). Pseudonymised data containing no personally-identifiable information were accessed 22nd June 2023.

### Data

**Religious/Spiritual Beliefs and Behaviours.** The RSBB variables used in this study are described in Table S1 of the Supporting Information, and include three broad measures of religiosity measured at two time-points (during pregnancy in 1991-1992 and approximately 5 years after delivery) in both the mothers and partners. These religiosity measures include: religious belief (via the question "Do you believe in God or some divine power?", with responses of 'yes', 'not sure' and 'no'), religious identity/affiliation (via the question "What sort of religious faith would you say you have?", coded as 'Christian' [of various denominations, but predominantly (~80%) Anglican/Church of England], 'other religion' [including a range of religions/faiths, such as Judaism, Islam, Hinduism, and 'other'] and 'None') and religious

attendance (via the question "Do you go to a place of worship?", with responses of 'at least once a week', 'at least once a month', 'at least once a year' and 'not at all'). These measures have been described in detail previously [40], and cover a range of theoretically-relevant religious beliefs and behaviours, including 'believing' (religious belief), 'behaving' (religious attendance) and 'belonging' (religious identity/affiliation) [29]. These – or very similar – questions have been widely used in previous religion research, both in ALSPAC [46,47] and more broadly in psychological, epidemiological and sociological disciplines [12,48,49]. We do note, however, that these variables may not cover the full range of religious beliefs, behaviours and experiences; we return to this important point in the discussion. Previous ALSPAC research has shown that, although RSBB is quite stable between these time-points, there is variation in these RSBBs over time, permitting their use in longitudinal analyses [39].

For all analyses, the RSBB exposure or outcome was explored using both the original coding of these variables (described above and in Table S1 of the Supporting Information), and also as dichotomised variables. Dichotomous variables were derived as follows: religious belief coded as 'yes' vs 'no' (with 'no' and 'not sure' responses combined together for 'no'), religious identity/affiliation coded as 'religious' vs 'other' (with 'Christian' and 'other religion' combined together for 'religious') and religious attendance coded as 'regular' vs 'occasional/non-attendance' (with 'at least once a week' and 'at least once a month' combined together for 'regular attendance' and 'at least once a year' and 'not at all' combined together for 'occasional/non-attendance'). We compared categorical and dichotomous RSBB variables because categorical variables on the original response scale avoid collapsing categories together, potentially resulting in a loss of information; however, binary variables – despite a potential loss of information – are arguably easier to interpret and allow comparisons with previous research which also often uses dichotomous RSBB outcomes (e.g., [11,12]).

**Depression and anxiety.** We used the depression and anxiety data collected in pregnancy, 2 years post-delivery, and 6 years post-delivery in both mothers and partners. Depression was assessed via the 10-item Edinburgh Postnatal Depression Scale (EPDS [50]), while anxiety was assessed via the anxiety subscale of the Crown-Crisp Experiential Index (CCEI-A [51]). Both scales had high internal validity (Cronbach's alpha > 0.80) and demonstrated construct validity when compared against other validated assessments of depression and anxiety in ALSPAC mothers (Center for Epidemiologic Studies Depression Scale for EPDS, and State and Trait subscales of the Spielberger State-Trait Anxiety Inventory for CCEI-A, respectively [38]).

When depression or anxiety were the exposures or outcomes of interest, we conducted analyses using both original continuous scores (0-30 for EPDS; 0-16 for CCEI-A) and binary markers to indicate probable depression and anxiety. For mothers, an EPDS score of 13 or more was used to define 'probable depression' [52], while for CCEI-A a score of 9 or more (corresponding to the 85th percentile of responses among ALSPAC mothers [38]) was used to define 'probable anxiety'. As mental health symptomology differs between men and women, different thresholds were used to indicate potential depression and anxiety in partners; an EPDS score of 10 or more was used to define 'probable depression' [53], while for CCEI-A a score of 6 or more (corresponding to the 85th percentile of responses among ALSPAC partners) was used to define 'probable anxiety'.

**Confounders.** In addition to baseline RSBB and mental health measured in pregnancy, we also adjusted for a wide range of covariates measured during pregnancy which may potentially cause both RSBB and mental health [5–7,54–56] and therefore act as potential confounders. This included: age, ethnicity, marital status, parity, various proxies of socioeconomic position (including highest educational attainment, area-level deprivation, household access to a car, occupational social class, recent financial difficulties, and home ownership status),

employment status, urban vs rural location, adverse childhood experiences, locus of control, inter-personal sensitivity measure of personality, subjective health status, body mass index (pre-pregnancy for mothers), physical activity levels, smoking status, alcohol intake, social networks and social support, and parental history of depression or anxiety. By adjusting for this wide range of baseline confounders, plus prior RSBB and mental health, we hope that this both rules out the possibility of reverse causality, while also making the assumption of no unmeasured confounding more plausible; if these conditions are met – which are impossible to verify – then a causal interpretation from longitudinal observational data may be warranted [13,14]. For full details of these confounders, see Table S2 in Supporting Information.

## Causal graphs

The causal graph in Fig 1 illustrates the hypothesised causal structure of the data when considering RSBB 5 years post-delivery as the exposure and mental health 6 years post-delivery as the outcome, while the causal graph in Fig 2 is for mental health 2 years post-delivery as the exposure and RSBB 5 years post-delivery as the outcome. Because the baseline confounders, RSBB and mental health variables were all measured at approximately the same time in pregnancy, the causal relations between these variables are not known with certainty, hence the bidirectional arrows for each of these variables indicating potential reciprocal causation. Even though these causal relations are uncertain, adjusting for all of these baseline covariates ought to remove as much confounding of the relationship between subsequent RSBB and mental health as possible. Note that three time-points of mental health data, but only two for RSBB, were needed to answer our research questions; that is, baseline mental health and RSBB in pregnancy (included as confounders in all analyses), mental health 2 years post-delivery as an exposure when RSBB 5 years post-delivery is the outcome (mental health 6 years post-delivery was not included in this analysis), and mental health 6 years post-delivery as the outcome when RSBB 5 years post-delivery was the exposure (mental health 2 years post-delivery was not included in this analysis).

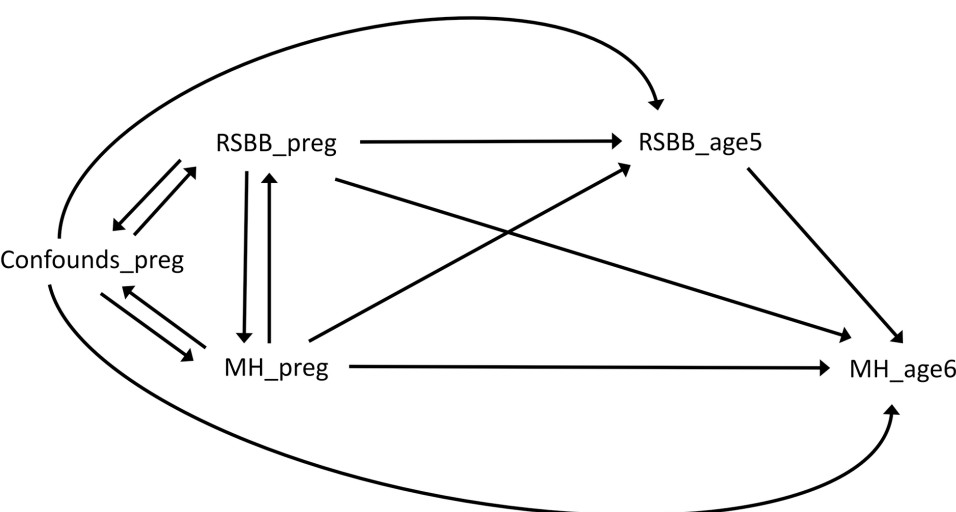

**Fig 1. Causal graph illustrating the assumed causal relations between variables when considering religious/spiritual beliefs and behaviours (RSBB) 5 years post-delivery as the exposure and mental health (MH) 6 years post-delivery as the outcome.** Note the bidirectional arrows between all baseline covariates in pregnancy; as these were all measured at approximately the same time the causal relations are not known with certainty.

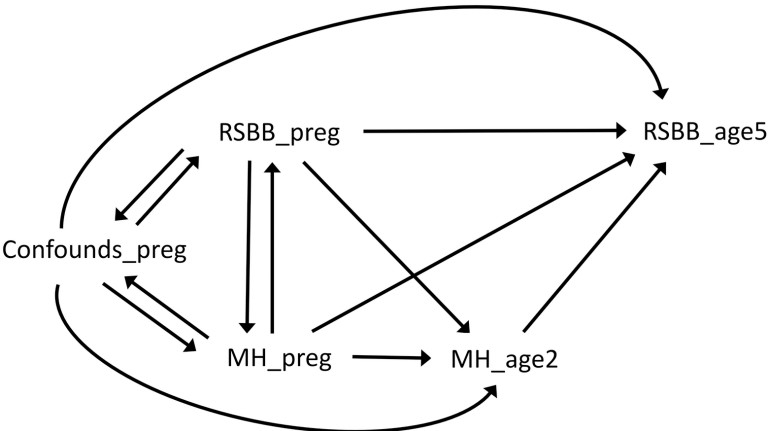

**Fig 2. Causal graph illustrating the assumed causal relations between variables when considering mental health (MH) 2 years post-delivery as the exposure and religious/spiritual beliefs and behaviours (RSBB) 5 years post-delivery as the outcome.** Note the bidirectional arrows between all baseline covariates in pregnancy; as these were all measured at approximately the same time the causal relations are not known with certainty.

## Analysis

Our first set of analyses aimed to test whether RSBB may cause subsequent mental health problems. Our exposure was RSBB (either religious belief, affiliation or attendance) 5 years post-delivery and our outcome was mental health (either depression or anxiety) 6 years post-delivery. We compared all combinations of RSBB exposures and mental health outcomes in both unadjusted and adjusted models, where adjusted models controlled for all baseline confounders in Table 2, plus baseline RSBB and mental health. We first used linear regression with continuous EPDS (depression) or CCEI-A (anxiety) scores as the outcome, initially with categorical RSBB and then binary RSBB exposures. Results of these linear regressions will be displayed on both the original scale (i.e., differences in raw EPDS or CCEI-A scores between levels of the RSBB exposure) and in standardised units (i.e., differences in EPDS or CCEI-A standard deviation scores between levels of the RSBB exposure) to provide information on effect sizes and allow depression and anxiety to be assessed on the same scale. These analyses were then repeated using binary depression and anxiety variables via logistic regression; where relevant, we also present predicted probabilities of the outcome to aid interpretation of these logistic models and their effect sizes. Analyses were conducted first on mothers, and then repeated for partners. These models are summarised in Table S3 of the Supporting Information.

Our next set of analyses investigated whether mental health may cause RSBB, using depression or anxiety 2 years post-delivery as the exposure and RSBB (religious belief, affiliation or attendance) 5 years post-delivery as the outcome. As above, we compared all combinations of mental health exposures and RSBB outcomes in both unadjusted and adjusted models, where adjusted models controlled for all baseline confounders in Table S2 of the Supporting Information, plus baseline RSBB and mental health. We first used multinomial logistic regression with the original categorical RSBB variable codings (Table S1 in Supporting Information), initially with continuous mental health variables (both on the original and standardised scales) and then with binary mental health markers. These analyses were then repeated using binary RSBB outcomes via logistic regression. For both multinomial and logistic models, we used predicted probabilities to aid interpretation of effect sizes where relevant; that is, we reported absolute effect sizes (i.e., %-point differences between levels of the exposure) in addition to

relative effect sizes (i.e., odds ratios and relative risk ratios), as the former are easier to interpret [57]. Analyses were repeated on mothers, and then partners. These models are summarised in Table S4 of the Supporting Information.

By adjusting for baseline confounders, including baseline RSBB and mental health, we hope that this will significantly reduce the risk of unmeasured confounding, as it is less likely that an unmeasured confounder would cause the exposure and outcome through a route other than via the prior exposure and outcome [13]. For instance, potential unmeasured confounders, such as personality traits or genetics, have a lower risk of confounding subsequent RSBB and mental health if prior RSBB and mental health are adjusted for. However, the possibility of unmeasured confounding (where unmeasured variables cause the exposure and outcome) and/or residual confounding (where observed confounders are measured with some error) may still result in bias [14,58,59]. To explore how robust these results are to potential unmeasured/residual confounding (hereafter 'unmeasured confounding'), where associations are found we conducted E-value sensitivity analyses to explore the level of unmeasured confounding necessary to explain away any observed association [60,61].

To formally test for differences between estimates from mothers' and partners' models, we conducted an interaction analysis using the approximations given in [62].

Throughout this paper we avoid interpreting results as 'statistically significant' based on arbitrary $p$-value thresholds (e.g., $p < 0.05$); instead, $p$-values are treated as continuous measures of evidence against (or incompatibility with) the null hypothesis of no difference, which are interpreted in conjunction with effect sizes, the range of plausible effect estimates (i.e., 95% confidence/compatibility intervals) and E-values for potential unmeasured confounding [63–65].

All analyses were conducted using the statistical software R, version 4.3.1 [66]

## Results

### Participant descriptive statistics

In the full sample of mothers 5 years post-delivery, 19.6% did not believe in God/a divine power, 33.9% were not sure and 46.5% did believe. 81.4% identified as 'Christian', 3.1% as 'other religion', and 15.5% had no religious affiliation, while only 20.0% of mothers regularly attended a place of worship at least once a month. Levels of religiosity were lower for partners (religious belief; 31.2% no, 35.2% not sure and 33.6% yes: religious identity; 73.2% Christian, 2.9% other and 23.9% none: religious attendance: 15.0% regular attendance).

Two years post-delivery, the median EPDS score for mothers was 5 (interquartile range [IQR] = 2 to 8), with 9.9% of mothers reaching a threshold for probable depression. Regarding anxiety at this time-point, the median CCEI-A score was 3 (IQR = 1 to 5), with 10.2% of mothers reaching a threshold for probable anxiety diagnosis. Depression and anxiety scores at this age were lower for partners (median EPDS = 3 [IQR = 1 to 5]; median CCEI-A = 2 [IQR = 1 to 4]), with concomitant lower rates of probable depression (depression = 8.1%; anxiety = 11.8%). Mental health scores and rates of potential diagnoses were slightly higher 6 years post-delivery (mothers: median EPDS = 5 [IQR = 2 to 10]; probable depression = 13.5%; median CCEI-A = 5 [IQR = 2 to 7]; probable anxiety = 17.7%; partners: median EPDS = 3 [IQR = 1 to 7]; probable depression = 13.7%; median CCEI-A = 3 [IQR = 1 to 5]; probable anxiety = 22.8%).

Descriptive statistics for all variables, in both the full ALSPAC sample and the complete-case analyses, are in Tables S5 (mothers) and S6 (partners) of the Supporting Information. Compared to the full sample, participants with complete data were older, had higher

levels of educational attainment, from less-deprived neighbourhoods, less likely to have an 'other than White' ethnicity, slightly more religious, and had marginally lower depression and anxiety scores.

## Analysis 1: Whether RSBB may cause mental health

**Mothers.** In these analyses, there were 3,856 mothers with complete data on all RSBB exposures 5 years post-delivery, mental health outcomes 6 years post-delivery and baseline confounders measured in pregnancy (28.2% of full 13,678 sample). Descriptive statistics of the exposures and outcomes, and their cross-tabulations, can be found in Table S7 of the Supporting Information. These descriptive statistics suggest that RSBB may be associated with slightly better mental health, as depression and anxiety scores, and rates of probable diagnoses, were slightly lower among religious individuals. For instance, across all binary religious exposures, rates of probable depression and anxiety were approximately 2 to 3%-points lower among religious individuals.

Fig 3 presents the unadjusted and adjusted results of the analyses using categorical RSBB exposures and standardised continuous depression and anxiety score outcomes. Focusing on the adjusted results, these findings indicate that, although in most cases the point estimates suggest a potential protective association between religiosity and lower depression and anxiety scores, the 95% confidence intervals frequently cross the null. Effect sizes are also very small, with most results consistent with a reduction in mental health scores between 0 and 0.2 standard deviation units. Together, this means that a range of effect sizes are consistent with these results, ranging from null to a weak protective association.

We can also explore the level of unmeasured confounding necessary to make these point estimates null via E-value sensitivity analyses. Using the adjusted association between 'Christian religious affiliation' and depression as an example (standardised effect size = -0.086; standard error [SE] = 0.055), an unmeasured confounder which increases the risk of both the exposure and outcome by approximately 40% is needed to make this effect null (using appropriate conversions from continuous to binary outcomes). While difficult to evaluate objectively, this does not appear to be a particularly large degree of unmeasured confounding, especially as the 95% confidence interval already crosses the null, and any unmeasured confounding weaker than this E-value would still push the point estimate closer to null. For comparison, the E-value to make the point estimate for the adjusted 'Other religious affiliation' association with depression estimate null (standardised effect size = -0.205; SE = 0.107) is approximately 1.75, meaning that a larger amount of unmeasured confounding would be necessary to shift this result towards the null; however, the 95% confidence interval again already crosses the null, and the sample size for this category is much smaller ($n$ = 93), meaning there is greater uncertainty in this estimate.

Analyses using binary RSBB exposures with continuous mental health outcomes (Fig S1), categorical RSBB exposures with binary mental health outcomes (Fig S2) and binary RSBB exposures with binary mental health outcomes (Fig S3; all in the Supporting Information) provided a broadly-consistent pattern of results, with weakly-protective and borderline-null associations between RSBB and subsequent mental health. For instance, a religious identity was associated with lower odds of a probable depression diagnosis (Odds Ratio [OR] = 0.660, 95% confidence interval [CI] = 0.548 to 0.960, $p$ = 0.0279), corresponding to a reduction in the predicted probability of being diagnosed by 4.2%-points (95% CI = -8.1 to -0.1). Full results of all analyses can be found in Table S8, with predicted probabilities from logistic models with binary mental health outcomes displayed in Table S9 and Figs S4 and S5, of the Supporting Information.

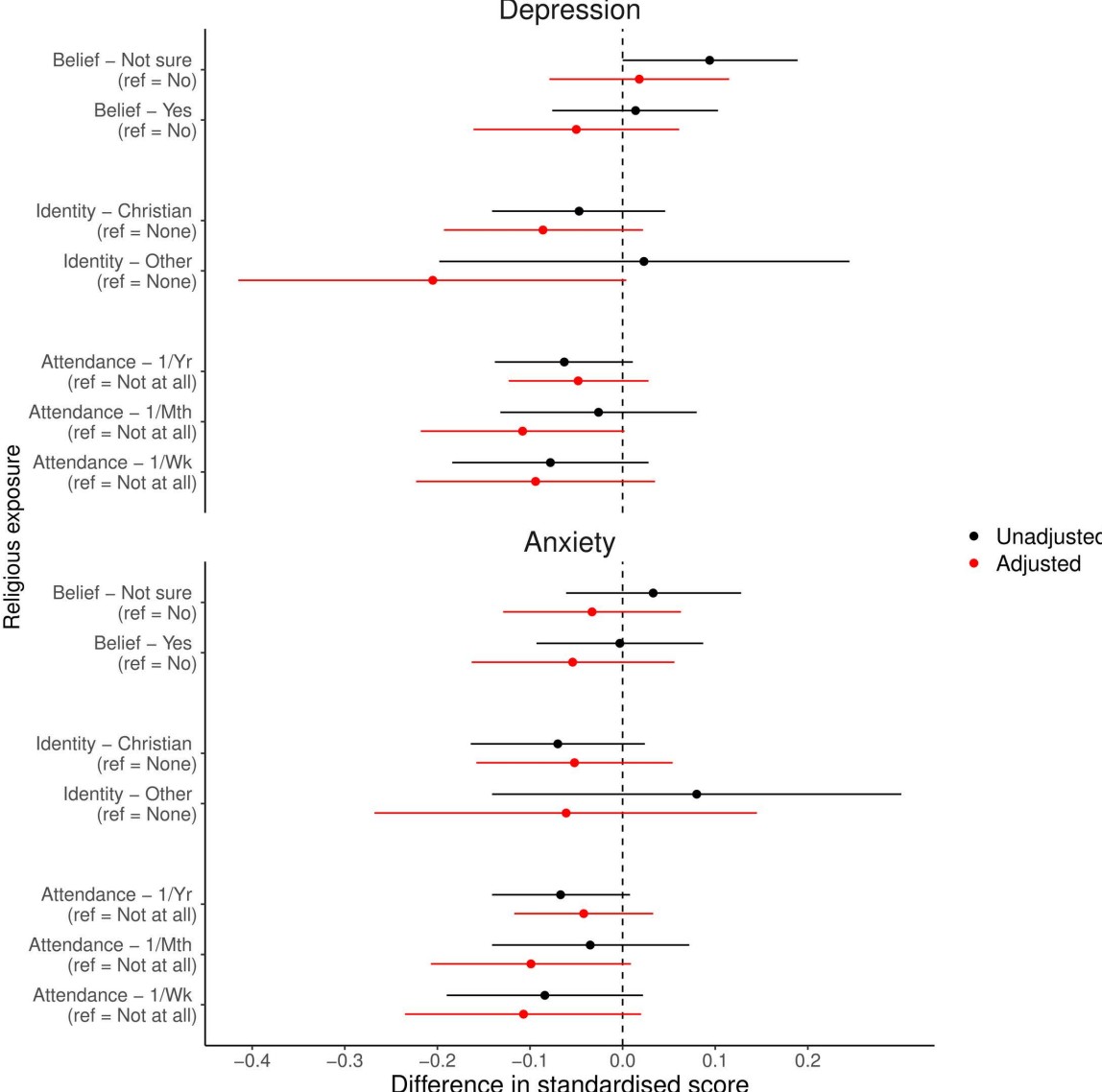

**Fig 3. Results of the mothers analyses with categorical religious/spiritual belief and behaviour (RSBB) exposures and standardised depression and anxiety scores as outcomes (n = 3,856).** Results in black are from unadjusted analyses, and those in red from adjusted analyses (adjusting for baseline confounders, RSBB and mental health). The dashed vertical line at '0' indicates a null association. Error bars denote 95% confidence intervals. Taking religious attendance, for instance, relative to the baseline category of 'not at all' attend, in adjusted analyses individuals who attended 'a minimum of once a week' were associated with a 0.09 unit (95% confidence interval between -0.22 and 0.04) decrease in standardised depression scores. Full results are in Table S8 of the Supporting Information.

**Partners.** For the equivalent analyses in partners, there were 1,940 individuals with complete data on all RSBB exposures 5 years post-delivery, mental health outcomes 6 years post-delivery and baseline confounders measured in pregnancy (14.6% of full 13,296 sample). Descriptive statistics of the exposures and outcomes, and their cross-tabulations, can be found in Table S10 of the Supporting Information. These descriptive statistics suggest that partners with a religious belief or affiliation may be associated with slightly worse mental health. For instance, rates of probable depression and anxiety were approximately 3 to 8%-points higher

among those with a religious belief or affiliation. Rates were even higher among those with an 'Other' religious affiliation (e.g., 40% with probable anxiety vs 18% for partners with no religious affiliation), but sample sizes for this 'other' religion category are small (*n* = 42 in total, 8 with probable depression and 17 with probable anxiety), meaning that these large differences should be interpreted with caution. Fewer differences in mental health were apparent for religious attendance.

Unadjusted and adjusted results of the analyses using categorical RSBB exposures and standardised continuous depression and anxiety score outcomes for partners are presented in Fig 4. Although the unadjusted results suggest a potential harmful association between religious

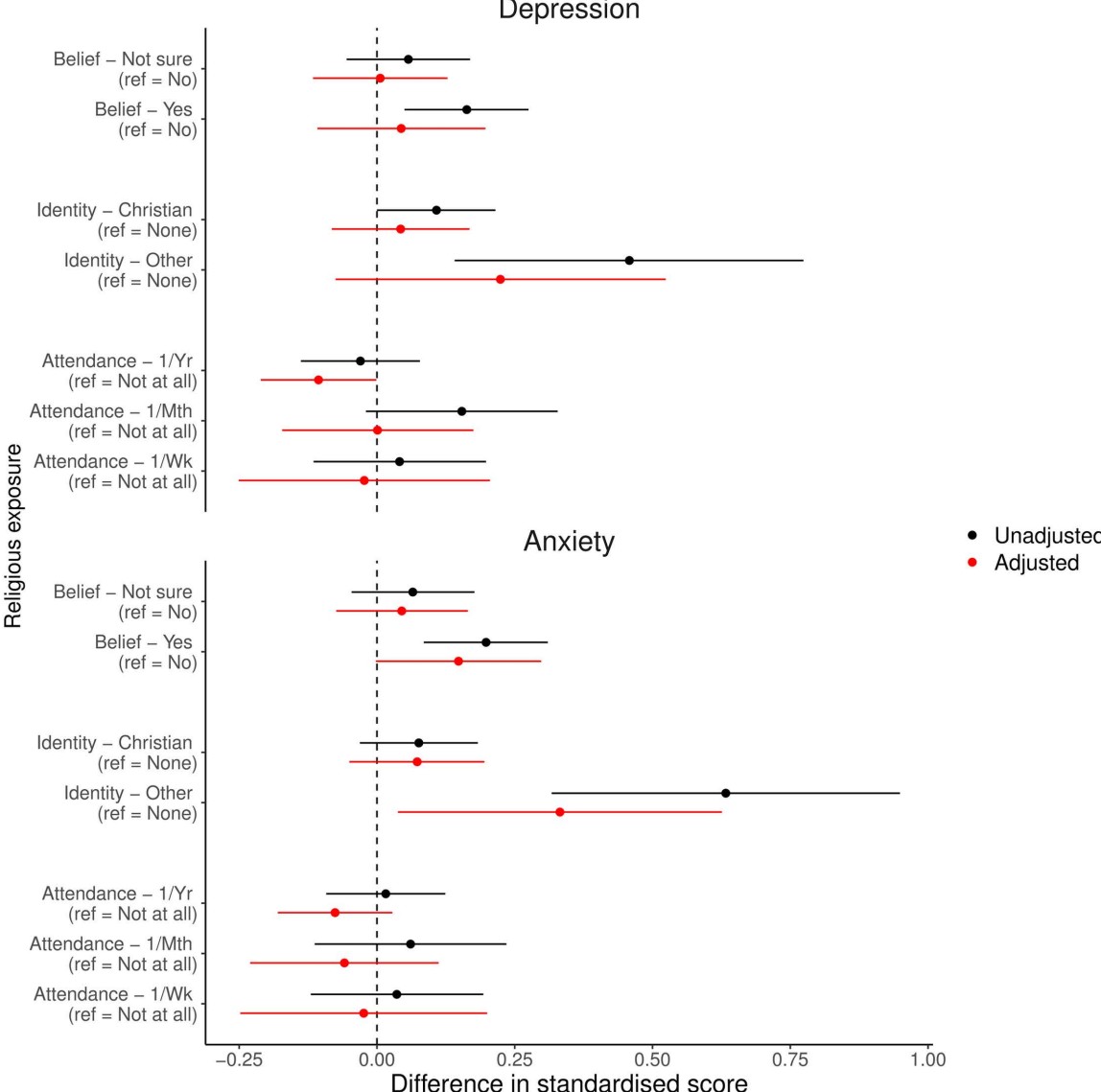

**Fig 4. Results of the partners analyses with categorical religious/spiritual belief and behaviour (RSBB) exposures and standardised depression and anxiety scores as outcomes (n = 1,940).** Results in black are from unadjusted analyses, and those in red from adjusted analyses (adjusting for baseline confounders, RSBB and mental health). The dashed vertical line at '0' indicates a null association. Error bars denote 95% confidence intervals. Full results are in Table S11 of the Supporting Information.

belief and affiliation and subsequent mental health, in adjusted analyses the majority of these associations are largely null. One exception is for religious belief and anxiety, with belief in God/a divine power associated with higher anxiety scores ($b$ = 0.148, 95% CI = -0.002 to 0.298, $p$ = 0.0526); the E-value to reduce this point estimate to null is ~ 1.6. A non-Christian religious identity was also associated with higher anxiety scores ($b$ = 0.332, 95% CI = 0.038 to 0.626, $p$ = 0.0270; with a similar, but weaker, association found for depression); the E-value to reduce this point estimate to null is ~ 1.6, while the E-value to shift the lower 95% confidence interval to null is ~ 1.25. These effect sizes are comparatively larger compared to the mothers' results, although again many of these partners' results are also consistent with either null or weak effects.

These partner results were similar when using different combinations of exposures and outcomes (binary RSBB exposures with continuous mental health outcomes in Fig S6; categorical RSBB exposures with binary mental health outcomes in Fig S7; binary RSBB exposures with binary mental health outcomes in Fig S8; all in the Supporting Information). For instance, religious belief was associated with approximately twice the odds of a probable anxiety diagnosis (OR = 1.926, 95% CI = 1.142 to 3.272, $p$ = 0.0146), corresponding to an increased in the predicted probability of being diagnosed by 7.9%-points (95% CI = 1.7 to 14.1). Full results of all partner analyses can be found in Table S11, with predicted probabilities from logistic models with binary mental health outcomes displayed in Table S12 and Figs S9 and S10, of the Supporting Information.

**Gender differences between mothers and partners.** Focusing on comparisons between mothers and partners with categorical RSBB exposures and standardised continuous mental health outcomes (Fig 5), there was little gender difference for religious attendance exposures. For religious belief and affiliation, however, some small differences were noted; for instance,

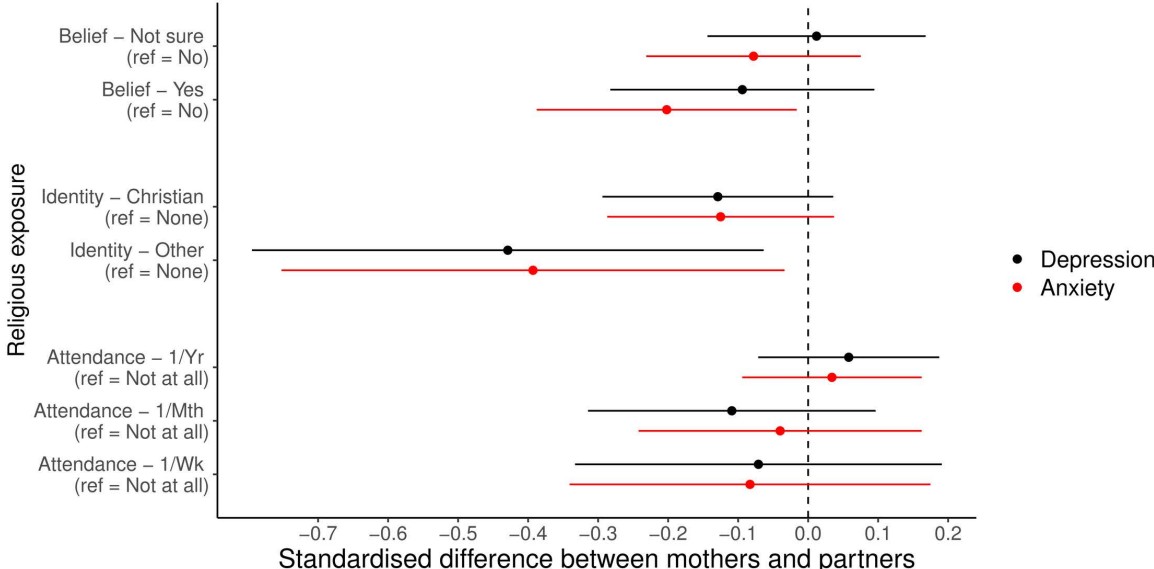

**Fig 5. Results of the interaction analyses assessing whether the adjusted mother and partner results differ, with categorical religious/spiritual belief and behaviour (RSBB) exposures and standardised depression and anxiety scores as outcomes.** Results in black are for the depression outcome, and those in red for anxiety. The dashed vertical line at '0' indicates no difference between mothers and partners, with results below 0 meaning that the estimate was lower in mothers, compared to partners. For instance, for 'Yes' to religious belief and anxiety (relative to answering 'No'), the mean difference in mothers was -0.05, while in partners it was 0.15, giving a standardised mean difference here of approximately -0.20. Error bars denote 95% confidence intervals. Full results are in Table S13 of the Supporting Information.

the association of an other religious identity with mental health scores in mothers was approximately a 0.4-standardised-units lower, compared to partners (depression: standardised difference = -0.429, 95% CI = -0.780 to -0.064, *p* = 0.0214; anxiety: standardised difference = -0.393, 95% CI = -0.752 to -0.034, *p* = 0.0320). The association between religious belief and anxiety scores was also approximately 0.2-standardised-units lower in mothers compared to partners (standardised difference = -0.202, 95% CI = -0.388 to -0.016, *p* = 0.0330), although this interaction was attenuated for depression (standardised difference = -0.094, 95% CI = -0.283 to 0.095, *p* = 0.3287). Similar patterns were observed for binary RSBB and continuous mental health (Fig S11), categorical RSBB and binary mental health (Fig S12), and binary RSBB and binary mental health (Fig S13; full results in Tables S13 [for continuous outcomes] and S14 [for binary outcomes]), reported in the Supporting Information.

## Analysis 2: Whether mental health may cause RSBB

**Mothers.**  The next set of analyses assessed whether mental health 2 years post-delivery may cause RSBB 5 years post-delivery. There were 4,025 mothers with complete data on all mental health exposures, RSBB outcomes and baseline pregnancy confounders (29.4% of full 13,678 sample). Descriptive statistics of the exposures and outcomes, and their cross-tabulations, can be found in Table S15 of the Supporting Information. These descriptive statistics suggest that mothers with worse mental health may be associated with lower rates of religiosity. For instance, mothers with a probable depression diagnosis were approximately 6%-points less likely to believe in God/a divine power, 5%-points more likely to have no religious affiliation, and 10%-points more likely to never attend a place of worship.

Fig 6 displays the predicted probabilities for each categorical RSBB outcome for a one-standardised-unit increase in either depression or mental health scores, based on the related multinomial regression model (full multinomial regression results are in Table S16, with pre-dicted probabilities in Table S17, of the Supporting Information). These results indicate that, for an increase in either depression or anxiety, there is little difference in the predicted proba-bilities of either religious belief or affiliation. For religious attendance, however, an increase in both depression and anxiety scores is associated with an increased probability of either never attending (depression = 1.90%-points [95% CI = 0.42 to 3.38]; anxiety = 1.27%-points [95% CI = -0.22 to 2.76]) or attending at least once a week (depression = 1.19%-points [95% CI = 0.31 to 2.07]; anxiety = 1.07%-points [95% CI = 0.17 to 1.97]), with a concomitant decrease in the probability of attending at least once a year (depression = -2.66%-points [95% CI = -4.25 to -1.06]; anxiety = -1.89%-points [95% CI = -3.49 to -0.28]).

We next applied an E-value sensitivity analysis to assess the level of unmeasured confounding necessary to overturn these results, using standardised depression scores as the exposure and attending at least once a year as the outcome, with 'not at all' as the reference (relative risk ratio [RRR] = 0.845, 95% CI = 0.760 to 0.939, *p* = 0.0018). This analysis suggested that an unmea-sured confounder which increased the risk of both the exposure and outcome by approximately 40% would be required to completely remove any association, while an approximately 20% increase in the risk of both would shift the 95% confidence interval to include the null. While subjective, this suggests that moderate levels of unmeasured confounders could overturn this result. Similar results were found using standardised anxiety scores (RRR = 0.890, 95% CI = 0.801 to 0.990, *p* = 0.0309), but with lower levels of unmeasured confounding necessary to over-turn results (E-value from RRR to null = 1.31; E-value from upper 95% CI to null = 1.08).

Results were similar when using different combinations of exposures and outcomes (continuous mental health exposures and binary RSBB outcomes in Fig S14; binary mental health exposures and categorical RSBB outcomes in Fig S15; binary mental health exposures and binary RSBB outcomes in Fig S16; see also Tables S16 and S17; all in the Supporting

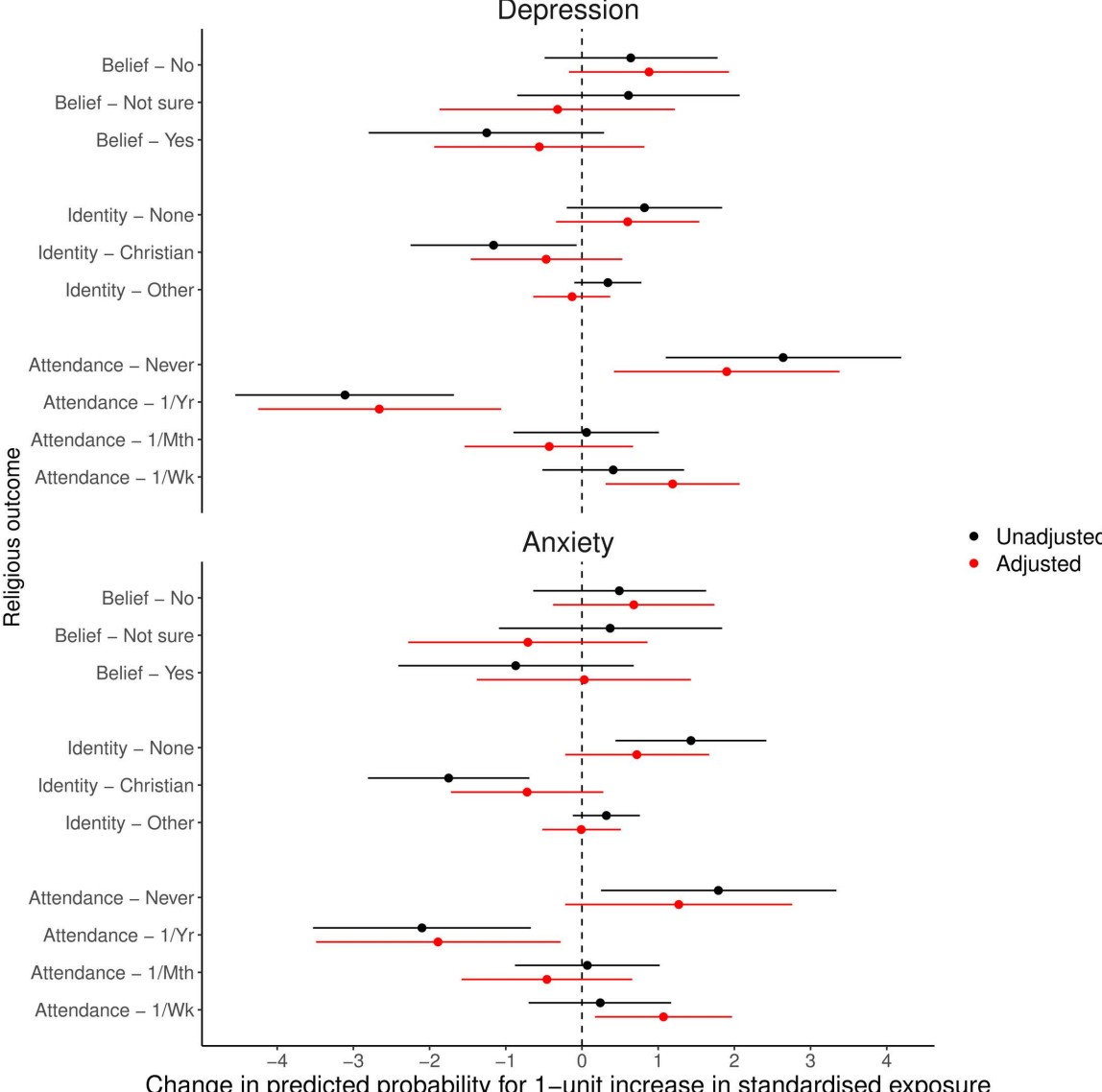

**Fig 6. Results of the mothers analyses with standardised depression and anxiety scores as exposures and categorical religious/spiritual belief and behaviour (RSBB) as outcomes (n = 4,025).** Results in black are from unadjusted analyses, and those in red from adjusted analyses (adjusting for baseline confounders, RSBB and mental health). This plot displays the predicted change in the probability of the RSBB outcome for a one-standardised-unit increase in the mental health exposure, based on the associated multinomial regression model. For instance, in adjusted analyses a one-standardised-unit increase in depression scores predicts a 1.90%-point increase in the probability of answering 'Never' for religious attendance (95% confidence interval = 0.42 to 3.38). The dashed vertical line at '0' indicates a null association. Error bars denote 95% confidence intervals. Full results of the multinomial models are in Table S16, with predicted probabilities in Table S17, of the Supporting Information.

Information). One difference when using binary RSBB outcomes is that associations with religious attendance disappear, because the previously-observed non-linear associations when using the categorical religious attendance outcome are obscured as categories are collapsed together.

**Partners.** Finally, we repeated the above analyses with mental health 2 years post-delivery as the exposure and RSBB 5 years post-delivery as the outcome in partners. 2,120 partners

had complete data on all exposures, outcomes and baseline confounders (15.9% of full 13,296 sample). Descriptive statistics of the exposures and outcomes, and their cross-tabulations, can be found in Table S18 of the Supporting Information. Unlike with mothers, these descriptive statistics suggest that partners with worse mental health may be associated with higher rates of subsequent religiosity. For instance, partners with a probable depression diagnosis were approximately 8%-points more likely to believe in God/a divine power, 8%-points less likely to have no religious affiliation, and 4%-points less likely to never attend a place of worship.

The predicted probabilities for each categorical RSBB outcome for a one-standardised-unit increase in either depression or mental health scores, based on the related multinomial regression model, are displayed in Fig 7 (full multinomial regression results are in Table S19, with predicted probabilities in Table S20, of the Supporting Information). Despite the unadjusted results suggesting a possible positive relationship between worse mental health and increased subsequent religiosity, in adjusted analyses the majority of associations were null. One potential exception is anxiety and religious attendance, with an increase in anxiety scores associated with a predicted increase in never attending by 1.66%-points (95% CI = -0.49 to 3.81) and a predicted decrease in attending at least once a year by 2.41%-points (95% CI = -4.59 to -0.23). However, to reduce the difference between attending at least once a year and not at all to null for standardised anxiety scores (RRR = 0.858, 95% CI = 0.735 to 1.003, *p* = 0.0546) requires relatively small levels of unmeasured confounding (E-value from RRR to null = 1.37), especially as the upper 95% confidence interval already crosses the null. Results were comparable when using different combinations of exposures and outcomes (continuous mental health exposures and binary RSBB outcomes in Fig S17; binary mental health exposures and categorical RSBB outcomes in Fig S18; binary mental health exposures and binary RSBB outcomes in Fig S19; see also Tables S19 and S20; all in the Supporting Information).

**Gender differences between mothers and partners.**  We first focus on comparisons between mothers and partners with standardised continuous mental health, or binary probable depression and anxiety diagnoses, exposures and categorical RSBB outcomes (Fig 8). For religious attendance outcomes, there were few differences by gender. For religious belief and affiliation, however, there was somewhat stronger evidence for differences between mothers and partners, although the evidence was relatively weak and many of the 95% confidence intervals included the null. For instance, a one-unit increase in standardised depression scores in mothers was associated with a lower relative risk of answering 'Yes' to religious belief (Ratio of RRRs = 0.814, 95% CI = 0.631 to 1.049, *p* = 0.1113) and answering 'Christian' to religious affiliation (Ratio of RRRs = 0.808, 95% CI = 0.643 to 1.016, *p* = 0.0678), relative to the baseline categories of 'No' and 'None', respectively, compared to partners. Similar patterns were observed for binary RSBB outcomes, although associations with religious belief were largely null in these analyses (Fig S20; full results in Tables S21 [for categorical outcomes] and S22 [for binary outcomes]; all in the Supporting Information).

## Discussion

In this paper we have made use of longitudinal religiosity and mental health data in a UK cohort study to examine whether there may be bidirectional causation between RSBB and mental health, and any potential differences by gender. We structure the discussion first in terms of whether the five predictions were met, followed by the implications of these findings, and then the strengths and limitations of this study.

*Prediction 1) RSBB will be protective against developing later depression and anxiety.* **Perhaps, but inconclusive (in mothers); No (in partners)**

In mothers the direction of the associations suggested a potential protective association, although effect sizes were small and uncertainty in the parameter estimates was consistent

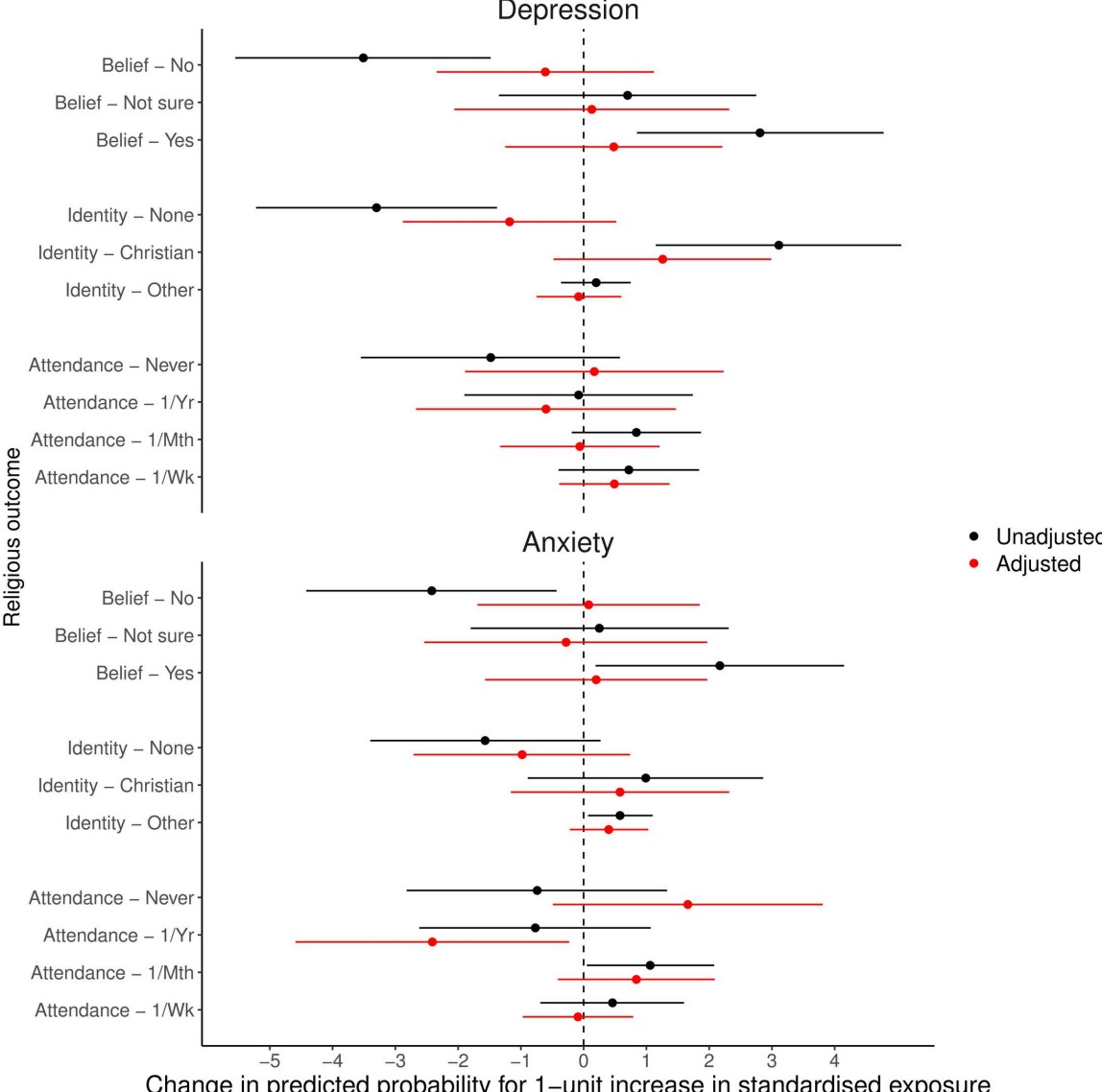

**Fig 7. Results of the partners analyses with standardised depression and anxiety scores as exposures and categorical religious/spiritual belief and behaviour (RSBB) as outcomes (n = 2,120).** Results in black are from unadjusted analyses, and those in red from adjusted analyses (adjusting for baseline confounders, RSBB and mental health). This plot displays the predicted change in the probability of the RSBB outcome for a one-standardised-unit increase in the mental health exposure, based on the associated multinomial regression model. For instance, in adjusted analyses a one-standardised-unit increase in depression scores predicts a 1.66%-point increase in the probability of answering 'Never' for religious attendance (95% confidence interval = -0.49 to 3.81). The dashed vertical line at '0' indicates a null association. Error bars denote 95% confidence intervals. Full results of the multinomial models are in Table S19, with predicted probabilities in Table S20, of the Supporting Information.

with plausible null associations. However, among partners, although many results were also consistent will the null, some associations with religious belief and affiliation suggested a potentially harmful (but weak) association with subsequent mental health. There was therefore no evidence that RSBB was a protective factor for mental health problems in partners.

*Prediction 2) Associations between RSBB and depression will be stronger than those between RSBB and anxiety:* **No**

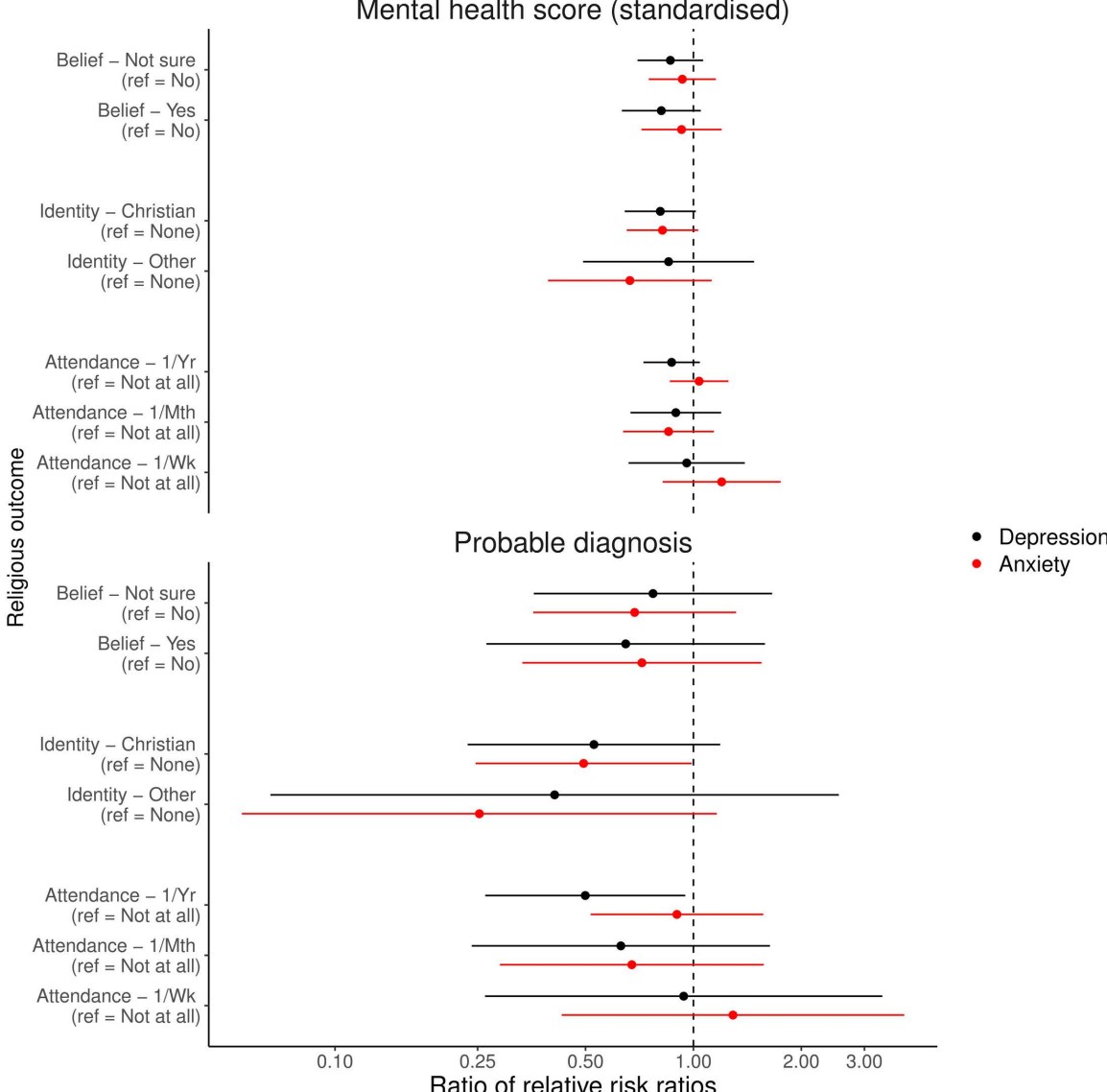

**Fig 8. Results of the interaction analyses assessing whether the adjusted mother and partner results differ, with standardised continuous mental health, or binary probable depression and anxiety diagnoses, as exposures and categorical religious/spiritual beliefs and behaviours (RSBB) as outcomes.** Results in black are for the depression outcome, and those in red for anxiety. The dashed vertical line at '1' indicates no difference between mothers and partners, with results below 1 meaning that the relative risk ratio (RRR) estimate was lower in mothers, compared to partners. For instance, for 'Christian' religious affiliation (relative to 'None') and continuous depression scores, a one-standardised-unit increase in depression was associated with a RRR of 0.92 in mothers, while in partners it was 1.14, giving a ratio of these RRRs here of approximately 0.81. Error bars denote 95% confidence intervals. Full results are in Table S21 of the Supporting Information.

Associations between RSBB and depression were of a comparable magnitude to those between RSBB and anxiety in mothers. In partners, while associations with both depression and anxiety were similar, the positive harmful association with RSBB may have been slightly stronger for anxiety, relative to depression. As depression and anxiety scores were highly correlated in this population ($r = \sim 0.7$), these lack of differences are perhaps not surprising.

*Prediction 3) Associations will be stronger for religious attendance, compared to other aspects of RSBB (i.e., religious belief or religious affiliation):* **No**

Associations were similar in magnitude for all RSBB exposures. Although small differences between RSBB exposures were noted – e.g., for partners' analyses, where religious belief and affiliation, but not attendance, were somewhat associated with higher anxiety scores – there was little evidence for a consistent direction of these effects.

*Prediction 4) Higher levels of depression and anxiety will be associated with lower rates of subsequent religious engagement and attendance:* **No**

There was little evidence that changes in depression or anxiety had any strong association with subsequent religious belief or affiliation, in either mothers or partners. Among mothers, higher depression and anxiety scores were associated with an increased probability of both never attending and attending at least once a week, suggesting a potentially more complex relationship between mental health and subsequent religiosity; however, these associations were only partially replicated in the partners, with higher anxiety (but not depression) scores associated with an increased probability of never attending and a decreased probability of attending at least once a year, but no difference in terms of attending once a week or once a month.

*Prediction 5) The above associations will be stronger among women compared to men:*
**Perhaps**

Some suggestive differences by gender were noted, such as RSBB having a potentially protective association with mental health in mothers and a potentially harmful association in partners, for instance. The formal comparison of mother and partner estimates partially supported this interpretation, with estimates lower in mothers relative to partners for religious belief and identity on subsequent mental health. However, in most cases the confidence intervals were also consistent with there being no differences, or differences being relatively small and/or uncertain. Some gender differences were also noted for the analysis of mental health on subsequent RSBB – with higher mental health scores more likely to be associated with a decrease in religious belief and affiliation (but not attendance) in mothers compared to partners – although again it is difficult to make firm conclusions as in many cases the results were consistent with either no or only small differences between mothers and partners.

In summary, the results reported here are somewhat mixed and inconclusive. There may be a potential protective association between RSBB and mental health issues in mothers, and a potential harmful association in partners, but effect sizes are weak and potentially consistent with there being no association. There is also little consistent evidence that mental health impacts subsequent religious belief or affiliation in mothers and partners, although it may impact religious attendance. Additionally, there may be small differences in these associations by gender, although the evidence is again inconclusive; if there are any differences, they are likely to be minor.

## Implications

Although much theoretical work [7,8], and some longitudinal empirical studies [12,23], have suggested that religion may be protective against subsequent mental health problems, we find little convincing evidence for such associations in this UK population. This is consistent with previous work indicating that such associations are highly heterogeneous between studies [22–24], although the reasons for this variation are unclear. Differences in religiosity between countries may be one potential explanation, with rates of religious belief and attendance lower in the UK, and other secularised Western European countries, compared to the US [27], where much previous work on religion and mental health has been conducted. Religion may plausibly have less of an impact on people's lives in more secular countries, and hence less impact on mental health. For instance, a meta-analysis on the relationship between RSBB and mental health in German-speaking areas found a much smaller protective association compared to

studies from the US [67]. However, other longitudinal work comparing a range of European countries with different rates of religiosity found similar null associations with subsequent depression [22], arguing against this interpretation. Additional work is needed to understand the reasons for this variation between studies, and this heterogeneity should be considered when thinking about implications for religion/spirituality-based mental health interventions [68], as their efficacy may be highly context-specific.

For the reverse direction – i.e., mental health impacting subsequent RSBB – we found little evidence that depression or anxiety may shape religious belief or affiliation. For religious attendance, however, we observed a potential non-linear association in ALSPAC mothers, with an increase in both depression and anxiety associated with both an increased probability of never attending *and* attending at least once a week. Assuming a causal interpretation, this suggests that mental health problems may drive people either way from or towards religion. Both directions are understandable, as – much like traumatic events [69] – worsening mental health problems could result in individuals either turning towards religion for comfort and support, or turning away from religion by questioning an individual's religious beliefs or withdrawing from social activities, including attending religious services [11]. However, this non-linear association was only partially-replicated in the partners cohort, where anxiety (but not depression) was associated with an increased probability of never attending, but with no difference in frequent attendance. Changes in mental health also seemed to have little association with subsequent religious belief or affiliation, suggesting that it may only impact religious attendance. However, this conclusion must remain tentative and requires replication in other studies, especially as relatively few previous studies have explored whether mental health impacts subsequent RSBB, and most have focused on religious attendance rather than other aspects of religiosity [11,12,28].

Finally, we observed – at best – weak evidence for gender differences in these relationships, with a somewhat protective association in mothers and a somewhat harmful association in their partners. However, these effect sizes were relatively small and often consistent with no difference, meaning the evidence for gender differences is suggestive but ultimately inclusive. If there *are* gender differences, this could perhaps be explained by different coping styles, with women more likely to use religion and social support (including religious social support) as coping mechanisms, compared to men [35]. Ultimately, more research is needed to understand first whether these gender differences are robust and replicable, and, second, what the potential mechanisms explaining this difference are.

## Strengths and limitations

This study has a number of key strengths, including: i) prospective longitudinal data, with baseline confounders, exposures and outcomes, from a largely-representative population-based birth cohort, helping to lower the risk of common biases such as confounding, reverse causality, selection bias and recall bias; ii) the use of validated depression and anxiety scales, providing support for construct validity in our mental health measurements; and iii) exploring associations with both depression and anxiety in the same sample, along with a range of different aspects of religiosity (i.e., belief, affiliation and attendance), permitting a more detailed exploration of the relations between these factors.

Despite these strengths, there are a number of limitations which may threaten both a causal interpretation of these associations, and also generalisability. First, while we adjusted for a range of baseline confounders, exposures and outcomes – hopefully minimising the risk of unmeasured confounding – we cannot rule out unmeasured confounders biasing our results. Our E-value sensitivity analyses assessed the level of such unmeasured confounding necessary to result in a null association, but as the majority of the 95% confidence intervals

already included the null these are difficult to interpret, and any unmeasured confounding could result in a weakening (or perhaps strengthening) of the results reported here. Relatedly, measurement error in either our confounders (i.e., residual confounding), or our exposures or outcomes could potentially result in bias [70]. While we again cannot rule this out, our use of a wide range of confounders for each key facet of interest (e.g., baseline depression and anxiety for prior mental health; baseline religious belief, identity and attendance for prior religiosity; a range of proxies for socio-economic position) should minimise somewhat the risk of residual confounding. Similarly, our use of validated mental health scales ought to reduce the risk of measurement error in these variables. While non-differential measurement error can result in bias towards the null, this is not always the case, and different types of measurement error can bias results either towards or away from the null [71]; the impact of measurement bias on our results – if any – is therefore difficult to establish with any degree of certainty.

Second, given that most of the 95% confidence intervals did cross the null, the majority of these results could be consistent with random variation and no (or rather small) associations. This may be especially possible for analyses which are underpowered due to small cell counts, such as analyses with binary mental health variables and 'other religion' affiliations; however, analyses with continuous mental health outcomes are likely to have greater power and less likely to be affected by random noise. The length of time between the exposures and outcomes – between 1 and 3 years – was relatively short. While comparable to many previous longitudinal studies in this area (e.g., [12,28]), it is possible that any effects of religion on mental health (or *vice versa*) may take longer to appear. We also note the issue of multiple comparisons here as there is a risk of some false positive associations due to the large numbers of analyses performed. However, we did not explicitly account for multiple comparisons because many of our tests were not independent from one another, meaning it is not clear how best to account for this (e.g., a Bonferroni adjustment would be overly-conservative). There is also conflicting advice regarding how best to adjust for multiple comparisons, if at all [72]. Given this, there is a risk of false positives here, but the extent of this is difficult to quantify and account for. Nonetheless, as most results were broadly consistent with null or small effect sizes, this risk is unlikely to alter our main conclusions.

A further potential issue is that of selection bias due to missing data and continued study participation [59,73]. Of the approximately 14,000 enrolled ALSPAC mothers and associated partners, only ~ 30% of mothers and ~ 15% of partners had complete data and were included in these analyses. Despite this large amount of missing data, we have focused on complete-case analyses here, rather than methods to impute missing data such as multiple imputation [74,75]. This is primarily because, although worse mental health has been associated with lower rates of ALSPAC participation [76], religious belief and affiliation are largely independent of ALSPAC participation when adjusting for sociodemographic factors [46]. In contrast, while religious attendance was still found to be associated with higher rates of participation, even when adjusting for sociodemographic variables [46], the magnitude of this effect is rather small and unlikely to result in meaningful bias in analyses [77]. Furthermore, there are no additional auxiliary variables to help predict the exposures or outcomes and provide additional external information to inform the imputations [78]; without any such informative auxiliary variables, multiple imputation is unlikely to reduce bias due to selection. Relatedly, as we adjusted for a range of baseline covariates, including RSBB and mental health, this may make the Missing-At-Random assumption more plausible (i.e., that missing data can be explained by the observed data). If this assumption is met, complete-case analyses will be unbiased – albeit somewhat inefficient (i.e., wider standard errors/confidence intervals [74]) – without the need for multiple imputation [79]. Despite the potential for selection bias, based on these considerations we expect the majority of our complete-case analyses to be relatively

unbiased, although perhaps inefficient. Although we believe our complete-case analyses to be largely unbiased, as suggested by a reviewer we performed multiple imputation to impute missing data as an additional sensitivity analysis; results were broadly consistent with the complete-cases results, albeit with somewhat greater precision (for more details, see section S2 of the Supporting Information, with results in Tables S23-S34 and Figs S21-S48).

As highlighted in the Methods section, our measurement of RSBB could be an additional potential limitation. While we explored a number of theoretically-relevant religion variables – religious belief, identity and attendance [29] – it is possible that other aspects of religiosity not explored here may have different associations with mental health, such as positive (or negative) religious coping, non-religious spirituality, intrinsic vs extrinsic religiosity, and private religious behaviours (e.g., prayer [24,67]). In addition, the focus on single questions to measure the traits explored may not fully capture these beliefs/behaviours compared to more detailed and validated multi-item scales, which could contribute to measurement error and bias. However, given our use of secondary ALSPAC data this was unfortunately unavoidable, as is common with many other longitudinal studies on this topic which focus on single-item measures of RSBB (e.g., [12,19,22,28]). A key avenue for future research would therefore be to embed a larger range of well-validated RSBB measures and scales within longitudinal studies to assess whether findings replicate using these more robust measures and further explore similarities and differences between different aspects of religiosity and their relations with mental health.

Generalisability may also be a concern. Both religion and mental health have changed considerably since these data were collected in the early 1990s, with religiosity declining [39,80] and increased societal acceptance, awareness and understanding of mental health issues. This makes it difficult to know how generalisable these results are to even the current UK population, let alone other populations with different levels of religious belief, different religions, and different approaches to mental health. Even within the UK, the ALSPAC sample is predominantly of White ethnicity and primarily Christian, making generalisations to other ethnicities and religious traditions challenging. Finally, as this sample focused on reproductive-age parents, specifically parents with young children, the extent to which these results are generalisable to other demographic groups – e.g., adults without children, adult with older children, adolescents, the elderly – is unclear.

## Conclusion

In conclusion, although some weak associations were reported, we found little evidence that RSBBs strongly impact subsequent mental health, or that mental health strongly shapes subsequent RSBBs. Some suggestive evidence for gender differences in these associations was reported – with a weak protective association between RSBB and mental health in mothers, but a weak harmful association in partners – although differences were relatively small and plausibly consistent with no differences. Given these mixed and inconclusive results, we cannot definitively say whether there is bidirectional causation between religion and mental health, or differences by gender, in this UK population but results are consistent with either null or small effect sizes.

## Supporting information

**S1 File. Supporting information for 'Exploring bidirectional causality between religion and mental health: A longitudinal study using data from the parent generation of a UK birth cohort'.** This supporting information file contains additional information regarding differences from the pre-registered analysis plan (Section S1) and information regarding

the multiple imputation sensitivity analysis (Section S2), in addition to all supporting tables (Tables S1-S34) and figures (Figs S1-S48).
(PDF)

## Acknowledgements

We are extremely grateful to all the families who took part in this study, the midwives for their help in recruiting them, and the whole ALSPAC team, which includes interviewers, computer and laboratory technicians, clerical workers, research scientists, volunteers, managers, receptionists and nurses.

## Author contributions

**Conceptualization:** Daniel Major-Smith, Isaac Halstead, Jean Golding.

**Data curation:** Daniel Major-Smith.

**Formal analysis:** Daniel Major-Smith.

**Funding acquisition:** Jean Golding.

**Methodology:** Daniel Major-Smith, Isaac Halstead, Jean Golding.

**Supervision:** Jean Golding.

**Visualization:** Daniel Major-Smith.

**Writing – original draft:** Daniel Major-Smith, Isaac Halstead, Jean Golding.

**Writing – review & editing:** Daniel Major-Smith, Jimmy Morgan, Isaac Halstead, Jean Golding.

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
