## [Decision Letter · Decision Letter 0]

3 Jan 2025

PONE-D-24-32137Exploring bidirectional causality between religion and mental health: A longitudinal study using data from the parent generation of a UK birth cohortPLOS ONE

Dear Dr. Major-Smith,

Thank you for submitting your manuscript to PLOS ONE. After careful consideration, we feel that it has merit but does not fully meet PLOS ONE’s publication criteria as it currently stands. Therefore, we invite you to submit a revised version of the manuscript that addresses the points raised during the review process.

We look forward to receiving your revised manuscript.

Kind regards,

Metin Çınaroğlu

Academic Editor

PLOS ONE

**Journal Requirements:**

The UK Medical Research Council and Wellcome Trust (Grant ref: 217065/Z/19/Z) and the University of Bristol currently provide core support for ALSPAC. This publication is the work of the authors, and Daniel Major-Smith will serve as guarantor for the contents of this paper. A comprehensive list of grants funding is available on the study website (http://www.bristol.ac.uk/alspac/external/documents/grant-acknowledgements.pdf). This project was made possible through the support of a grant from the John Templeton Foundation (ref no. 61917). The opinions expressed in this publication are those of the author(s) and do not necessarily reflect the views of the John Templeton Foundation.

Reviewers' comments:

Reviewer's Responses to Questions

**Comments to the Author**

1. Is the manuscript technically sound, and do the data support the conclusions?

Reviewer #1: Yes

Reviewer #2: No

Reviewer #3: Yes

2. Has the statistical analysis been performed appropriately and rigorously? 

Reviewer #1: Yes

Reviewer #2: I Don't Know

Reviewer #3: Yes

3. Have the authors made all data underlying the findings in their manuscript fully available?

Reviewer #1: Yes

Reviewer #2: No

Reviewer #3: Yes

4. Is the manuscript presented in an intelligible fashion and written in standard English?

Reviewer #1: Yes

Reviewer #2: No

Reviewer #3: Yes

5. Review Comments to the Author

**Reviewer #1:**  The article in question stands out for its well-organized structure and the use of an appropriate academic style, which facilitates the communication of the results and analyses carried out. Each section of the work—from the initial presentation to the data analysis and the subsequent discussion—is articulated clearly and coherently, allowing the reader to easily follow the logical flow of the discourse. The final images, carefully chosen and well integrated into the text, play a crucial role, as they make the results more accessible and immediately understandable, thereby contributing to a better assimilation of the information presented.

However, there are significant improvements that could be made to the introductory part. Currently, the initial overview could benefit from a broader context that situates the work within the existing research framework. Including references to previous studies and similar research would not only enrich the content but also allow for a clearer highlighting of how this specific study fits into the current scientific dialogue. An introduction that outlines the gaps in the previously existing literature and emphasizes the originality of the adopted approach would provide readers with a deeper understanding of the value and importance of the work, making the contribution even more significant. This contextual enrichment would not only serve to attract the reader's interest but also justify the relevance of the research as a whole.

**Reviewer #2:**  Dear authors. This started as a very interesting paper, but I can't seem to find any tables attached. Without tables, I am not able to see your findings, understanding your models, or make any recommendations. Therefore, I had to reject this iteration of the paper.

**Reviewer #3:**  "Exploring Bidirectional Causality between Religion and Mental Health: A Longitudinal Study Using Data from the Parent Generation of a UK Birth Cohort"

The study benefits from a robust dataset and advanced statistical approaches, such as pre-registered analyses and E-value sensitivity tests. However, the results were largely inconclusive, with small or null associations found between religiosity and mental health. While the study provides valuable insights, several methodological and conceptual issues limit its contributions.

Recommendations for Revision

1. Enhance Measurement Precision:

- Expand religiosity measures to include aspects like private prayer, intrinsic religiosity, and spiritual coping.

- Consider using diagnostic tools or broader measures of mental health to improve outcome validity.

2. Address Missing Data:

- Implement multiple imputation or sensitivity analyses to better address attrition bias.

- Provide a detailed comparison of participants with and without complete data to clarify potential biases.

3. Examine Nonlinear Relationships:

- The findings suggest potential non-linear associations (e.g., mental health influencing both increased and decreased religious attendance). Future analyses should formally test for such patterns.

4. Discuss Limitations More Transparently:

- Acknowledge the limitations of generalizing findings to non-secular populations.

- Discuss the potential impact of residual confounding and measurement error on null results.

5. Strengthen Interpretative Framework:

- Delve deeper into the implications of gender-specific differences and why religious belief may show protective associations in women but harmful effects in men.

- Explore alternative theoretical frameworks, such as coping theories, that may explain bidirectional relationships more robustly.

6. Condense Statistical Presentation:

- While detailed statistical tables are helpful, consider summarizing key findings in the main text to improve readability and focus on meaningful results.

Conclusion

This manuscript tackles an important and underexplored topic but falls short in delivering conclusive or actionable insights due to methodological and interpretive gaps. Addressing the recommendations above will enhance the study’s clarity, rigor, and relevance to both academic and clinical audiences. Despite its limitations, the study contributes to the growing body of research on religion and mental health, particularly in secular contexts.

6. PLOS authors have the option to publish the peer review history of their article (what does this mean? ). If published, this will include your full peer review and any attached files.

**Do you want your identity to be public for this peer review?** For information about this choice, including consent withdrawal, please see our Privacy Policy .

Reviewer #1: No

Reviewer #2: No

Reviewer #3: No

---

## [Author Response · Author response to Decision Letter 1]

20 Jan 2025

Please see the 'RSBBandParentalMH_ResponseToReviewers_rev1.docx' file submitted with the rest of the manuscript files.

---

## [Decision Letter · Decision Letter 1]

10 Feb 2025

Exploring bidirectional causality between religion and mental health: A longitudinal study using data from the parent generation of a UK birth cohort

PONE-D-24-32137R1

Dear Authors,

We’re pleased to inform you that your manuscript has been judged scientifically suitable for publication and will be formally accepted for publication once it meets all outstanding technical requirements.

Kind regards,

Metin Çınaroğlu

Academic Editor

PLOS ONE

Additional Editor Comments (optional):

Reviewers' comments:

Reviewer's Responses to Questions

**Comments to the Author**

1. If the authors have adequately addressed your comments raised in a previous round of review and you feel that this manuscript is now acceptable for publication, you may indicate that here to bypass the “Comments to the Author” section, enter your conflict of interest statement in the “Confidential to Editor” section, and submit your "Accept" recommendation.

Reviewer #1: (No Response)

Reviewer #3: All comments have been addressed

2. Is the manuscript technically sound, and do the data support the conclusions?

Reviewer #1: Yes

Reviewer #3: Yes

3. Has the statistical analysis been performed appropriately and rigorously? 

Reviewer #1: Yes

Reviewer #3: Yes

4. Have the authors made all data underlying the findings in their manuscript fully available?

Reviewer #1: (No Response)

Reviewer #3: Yes

5. Is the manuscript presented in an intelligible fashion and written in standard English?

Reviewer #1: (No Response)

Reviewer #3: Yes

6. Review Comments to the Author

Reviewer #1: The article after previous reviews conducted in my opinion seems to be able to be published in the form presented.

Reviewer #3: All the comments have been addressed in the revision, so no need to write more review comments in this section

7. PLOS authors have the option to publish the peer review history of their article (what does this mean? ). If published, this will include your full peer review and any attached files.

**Do you want your identity to be public for this peer review?** For information about this choice, including consent withdrawal, please see our Privacy Policy .

Reviewer #1: No

Reviewer #3: No

---

## [Editor Report · Acceptance letter]

PONE-D-24-32137R1

PLOS ONE

Dear Dr. Major-Smith,

I'm pleased to inform you that your manuscript has been deemed suitable for publication in PLOS ONE. Congratulations! Your manuscript is now being handed over to our production team.

Kind regards,

on behalf of

Dr. Metin Çınaroğlu

Academic Editor

PLOS ONE